# Markers of prolonged hospitalisation in severe dengue

**Mario Recker** [ID][1,2☯] *, **Wim A. Fleischmann**[1☯], **Trinh Huu Nghia**[3], **Nguyen Van Truong**[3], **Le Van Nam**[3], **Do Duc Anh**[1,4], **Le Huu Song**[4,5], **Nguyen Trong The**[4,5], **Chu Xuan Anh**[5], **Nguyen Viet Hoang**[5], **Nhat My Truong**[4,5], **Nguyen Linh Toan**[4,6], **Peter G. Kremsner**[1,7], **Thirumalaisamy P. Velavan**[1,4,8]

**1** Institute for Tropical Medicine, University Hospital Tübingen, Tübingen, Germany, **2** Centre for Ecology and Conservation, University of Exeter, Penryn Campus, Penryn, United Kingdom, **3** 103 Military Hospital, Vietnam Military Medical University, Hanoi, Vietnam, **4** Vietnamese-German Center for Medical Research, VG-CARE, Hanoi, Vietnam, **5** 108 Military Central Hospital, Hanoi, Vietnam, **6** Department of Pathophysiology, Vietnam Military Medical University, Hanoi, Vietnam, **7** Centre de Recherches Médicales de Lambaréné (CERMEL), Gabon, **8** Faculty of Medicine, Duy Tan University, Da Nang, Vietnam

☯ These authors contributed equally to this work.
* mario.recker@uni-tuebingen.de

## Abstract

### Background

Dengue is one of the most common diseases in the tropics and subtropics. Whilst mortality is a rare event when adequate supportive care can be provided, a large number of patients get hospitalised with dengue every year that places a heavy burden on local health systems. A better understanding of the support required at the time of hospitalisation is therefore of critical importance for healthcare planning, especially when resources are limited during major outbreaks.

### Methods

Here we performed a retrospective analysis of clinical data from over 1500 individuals hospitalised with dengue in Vietnam between 2017 and 2019. Using a broad panel of potential biomarkers, we sought to evaluate robust predictors of prolonged hospitalisation periods.

### Results

Our analyses revealed a lead-time bias, whereby early admission to hospital correlates with longer hospital stays - irrespective of disease severity. Importantly, taking into account the symptom duration prior to hospitalisation significantly affects observed associations between hospitalisation length and previously reported risk markers of prolonged stays, which themselves showed marked inter-annual variations. Once corrected for symptom duration, age, temperature at admission and elevated neutrophil-to-lymphocyte ratio were found predictive of longer hospitalisation periods.

---

**Data Availability Statement:** Data analysed in this work is provided as Supplementary Material.

**Funding:** TPV, LHS, NLT, NTT, NXH, and NMT acknowledge the PAN-ASEAN Coalition for Epidemic and Outbreak Preparedness (PACE-UP)

(DAAD Project ID: 57592343). The funders had no role in study design, data collection and analysis, decision to publish, or preparation of the manuscript.

**Competing interests:** The authors have declared that no competing interests exist.

## Conclusion

This study demonstrates that the time since dengue symptom onset is one of the most significant predictors for the length of hospital stays, independent of the assigned severity score. Pre-hospital symptom durations need to be accounted for to evaluate clinically relevant biomarkers of dengue hospitalisation trajectories.

### Author summary

Dengue places a significant burden on healthcare settings. Especially in low and middle income settings and during large outbreaks, allocation of limited resources to those at high risk of morbidity and mortality can be critically important. Various risk factors of severe infection outcomes and hospitalisation, such as secondary heterologous infection, have been described, yet reliable biomarkers predictive of prolonged stays once hospitalised are still lacking. In this work we analysed dengue hospitalisation data collected over a period of three consecutive years in Northern Vietnam, which revealed an unexpected negative correlation between dengue severity and length of hospitalisation. Further analysis showed that this was primarily driven by a longer period between symptom onset and admission in those patients with a higher severity score. Moreover, we found that this delay negated other observed correlates of prolonged hospital stays, which themselves revealed significant inter-annual variations. Taken together, this work demonstrates that time to admission is one of the strongest predictors of hospitalisation length and that this needs to be taken into consideration for finding reliable biomarkers of predicted healthcare needs in patients admitted to hospital due to dengue.

## Background

Dengue is a mosquito-borne viral disease caused by infection with any of the four antigenically distinct serotypes of the dengue virus (DENV). Dengue is highly prevalent in many tropical and sub-tropical regions around the world, placing a significant burden on public health systems and local economies. Incidence geographical distribution of dengue has increased significantly in recent decades [1,2], with an estimated 400 million infections annually, of which around 96 million present clinical symptoms [3]. The outcome of an infection is highly variable, ranging from asymptomatic infections to life-threatening disease, with the most important risk factor for the development of severe disease being a secondary, heterotypic infection through the phenomenon of antibody-dependent enhancement [4,5].

Despite low mortality rates, a large number of patients get hospitalised with dengue each year, which poses a considerable economic burden on local health systems, especially in low- and middle-income countries (LMICs) [6]. The length of hospital stay, and thus medical attention required, can vary from a few days to weeks. Currently, besides prophylactic measures there is no licensed anti-viral therapy, and treatment is limited to supportive clinical care, including management of fever and fluid balance. It is not yet clear to what degree supportive care can directly influence the immunopathology of dengue and how this in turn affects the clinical course of the infection.

To date, extensive scientific research has concentrated on identifying immunopathologic markers associated with or predictive of the severity of an infection (see e.g. [7–15]). On the

other hand, only a few studies have focused on hospitalisation length and factors associated with this [16–19]. Although these have put forward a wide variety of predictive markers, such as aspartate-aminotransferase (AST) or platelets [20], no clear consensus on their associations with hospitalisation length has yet been reached [8,17]. An added complication is that the length of a hospital stay may be affected by external factors related to health resource requirements and availability, which themselves can vary from year to year. That is, dengue's complex epidemiology is characterised by multi-annual oscillations and replacement of dominant serotypes, both of which have a direct influence on disease incidence and thus frequency of hospital admissions. In LMICs, these epidemiological changes have implications for the medical care of patients during major and prolonged outbreaks when health resources are scarce.

In this study we analysed dengue hospitalisation data collected in Vietnam over a three-year period to improve our understanding of risk factors associated with prolonged hospital stays. Our analysis reveals an unexpected correlation between markers of disease severity and hospitalisation length, which can be resolved by taking the period of dengue symptoms prior to hospitalisation into account. These results may have important health economic implications in Vietnam and other LMICs where dengue is endemic.

## Methods

### Data

A total of 1852 patients admitted to the 103 Military Central Hospital, Viet Nam Military Medical University and the 108 Military Central Hospital, Hanoi, Vietnam, between 2017 and 2019 with a recorded dengue severity score at the time of admission and a hospital stay $\geq$ 1 day were evaluated. For each patient, up to 47 variables were collected at admission and during the course of hospitalisation (see S1 Table). Peripheral blood was taken from the first until the eighth day post admission, although here we analysed only the very first measure with respect to a number of biomarkers associated with disease severity score (1, 2, or 3; see below for details) or hospitalisation length. Despite the availability of different variables and repeated measurements taken during hospital stay, due to a high degree of missingness (i.e. incomplete records) we restricted our analysis to the following list of demographic and diagnostic data: age, temperature on admission, gender, blood group, pulse rate, blood pressure, disease symptoms (such as headache, rash, bleeding, etc), days since symptom onset (here referred to as *symptom days*), length of hospital stay, disease severity, and blood values (incl. leukocytes (WBC), lymphocytes (LYM), erythrocytes (RBC), haemoglobin (Hb), haematocrit (HCT), platelets (PLT), and neutrophils (NEU)). Note, even though many of the blood markers considered here showed a high degree of missingness, their numbers were still sufficient to derive statistically robust inferences.

### Data selection for analysis

Of the 1852 patients, 5 died (2017: n = 1; 2019: n = 4); these were excluded from the analysis due to the lack of additional clinical and meta data. We further excluded patients with recorded hospital stays of >21 days as well as those with a recorded severity score of 3. Finally, we omitted patients whose severity score changed during the course of hospitalisation, resulting in a total of N = 1593 patients for downstream analysis. The reason for excluding individuals with a severity score of 3 is that the numbers were too small to derive statistically meaningful inferences, and the exclusion of individuals with hospital stays >21 days was due to the distribution of hospital stays showing a sharp drop off after 10 days, resulting in 95% of all records having a recorded hospital stay of <21 days, with the remaining 5% being almost uniformly distributed between 21 and 300 days.

### Clinical presentation

In suspected cases of dengue infection, patients undergo a rapid dengue test consisting of NS1 antigen, IgM and IgG rapid test. The final decision on admission is made considering the patient's condition and the results of the rapid test. The patient's condition depends on the co-existing diseases, social situation and severity of the disease.

The severity of the disease is graded in a three-level scale according to the WHO "Guidelines for Diagnosis, Treatment, Prevention and Control of Dengue Fever", 2009 edition, released for use by the Vietnamese Ministry of Health in July 2023. The three grades [1 to 3] of severity of dengue infection are: dengue fever without warning signs (severity 1), dengue fever with warning signs (severity 2), and severe dengue fever (severity 3). These are assigned according to the following criteria.

Dengue without warning signs (severity 1): living or travelled to the endemic areas, fever within 7 days, and at least two of the following signs: nausea/vomiting, skin rash, joint and muscle pain, positive tourniquet test, thrombocytopenia, NS1 (+) or dengue IgM (+).

Dengue with warning signs (severity 2): diagnosed with dengue infection and having one of the following signs: abdominal pain or tenderness, persistent vomiting, fluid accumulation (ascites or pleural effusion), mucosal bleeding, lethargy, restlessness, liver enlargement greater than 2 cm. Another laboratory criterion considered as a warning sign is an increase in haematocrit accompanied by a rapid decrease in platelet count.

Severe dengue cases (severity 3): diagnosed with dengue infection and having one of the following signs: severe plasma loss leading to shock, which could lead to shock or respiratory failure (due to fluid accumulation), severe bleeding, severe organ failure (incl. liver failure (AST or ALT $> = 1000$ U/L), renal failure (creatinine $>$ upper limit of normal range), coma (mental disorder), myocarditis / heart failure, other organ failures).

Note, in this work we refer to dengue severity exclusively in the context of the assigned severity score at admission; patients whose severity score was changed during their hospital stay were not considered in this analysis.

### Statistical analysis

Unless stated otherwise, associations between severity scores and biomarkers were assessed using 2-sided t-tests. The effects of measured biomarkers on length of hospital stay were inferred using generalised linear models (GLM) with a Poisson error structure. All statistical analyses were performed in R version 4.2 (www.R-project.org).

## Results

### Patient characteristics on admission

A total of 1593 hospital admitted patients with a diagnosis of dengue and a severity score between 1 and 2 according to WHO definitions were analysed (see Methods). Table 1 provides an overview of the data, stratified by disease severity score. (S1 Table provides an overview of the entire dataset, including clinical variables not considered here and of those patients with severity score 3).

The most common symptoms on admission were headache (98%) and body ache (97%). Fever was seen in around two thirds of patients on admission, although the vast majority of individuals (96%) reported a fever episode before admission. Signs of bleeding, such as nose, muscle or gum but also including positive Tourniquet test, petechia and purpura, was reported in 33% of patients and, as expected, was positively correlated with severity score (21% vs 76%

**Table 1. Overview of patient characteristics.**

| | Severity 1 (N = 1238) | Severity 2 (N = 355) | Total (N = 1593) | p value |
|---|---|---|---|---|
| **Year** | | | | 0.023 |
| 2017 | 935 (75.5%) | 258 (72.7%) | 1193 (74.9%) | |
| 2018 | 162 (13.1%) | 38 (10.7%) | 200 (12.6%) | |
| 2019 | 141 (11.4%) | 59 (16.6%) | 200 (12.6%) | |
| **Age** | | | | < 0.001 |
| Mean (CI) | 39.6 (38.7, 40.5) | 35.5 (34.1, 37.0) | 38.7 (37.9, 39.5) | |
| Range (Min - Max) | 13.0–91.0 | 13.0–78.0 | 13.0–91.0 | |
| **Sex** | | | | 0.206 |
| female | 605 (48.9%) | 187 (52.7%) | 792 (49.7%) | |
| male | 633 (51.1%) | 168 (47.3%) | 801 (50.3%) | |
| **Day of illness** | | | | < 0.001 |
| Mean (CI) | 3.5 (3.4, 3.6) | 4.8 (4.7, 5.0) | 3.8 (3.7, 3.9) | |
| Range (Min - Max) | 1.0–10.0 | 1.0–8.0 | 1.0–10.0 | |
| **Hospital stay [days]** | | | | < 0.001 |
| Mean (CI) | 5.6 (5.4, 5.7) | 4.8 (4.6, 5.0) | 5.4 (5.3, 5.5) | |
| Range (Min - Max) | 1.0–18.0 | 1.0–15.0 | 1.0–18.0 | |
| **Pulse** | | | | < 0.001 |
| Mean (CI) | 90.2 (89.5, 90.9) | 86.12 (85.0, 87.3) | 89.2 (88.6, 89.9) | |
| Range (Min - Max) | 55.0–150.0 | 56.0–120.0 | 0.0–150.0 | |
| Missing | 15 | 1 | 16 | |
| **Temperature** | | | | < 0.001 |
| Mean (CI) | 38.2 (38.2, 38.3) | 37.9 (37.8, 38.0) | 38.2 (38.1, 38.2) | |
| Range (Min - Max) | 35.0–40.6 | 35.2–41.0 | 35.0–41.0 | |
| Missing | 31 | 5 | 36 | |
| **Systolic blood pressure** | | | | < 0.001 |
| Mean (CI) | 114.5 (113.6, 115.3) | 111.0 (109.5, 112.4) | 113.7 (112.9, 114.4) | |
| Range (Min - Max) | 80.0–181.0 | 80.0–170.0 | 80.0–181.0 | |
| Missing | 15 | 0 | 15 | |
| **Diastolic blood pressure** | | | | 0.087 |
| Mean (CI) | 70.8 (70.3, 71.4) | 69.8 (68.8, 70.8) | 70.6 (70.1, 71.1) | |
| Range (Min - Max) | 50.0–130.0 | 40.0–100.0 | 40.0–130.0 | |
| Missing | 15 | 0 | 15 | |
| **Headache** | | | | 0.957 |
| no | 27 (2.2%) | 8 (2.3%) | 35 (2.2%) | |
| yes | 1197 (97.8%) | 347 (97.7%) | 1544 (97.8%) | |
| Missing | 14 | 0 | 14 | |
| **Body ache** | | | | 0.732 |
| no | 37 (3.0%) | 12 (3.4%) | 49 (3.1%) | |
| yes | 1187 (97.0%) | 343 (96.6%) | 1530 (96.9%) | |
| Missing | 14 | 0 | 14 | |
| **Fatigue** | | | | 0.026 |
| no | 346 (28.0%) | 121 (34.1%) | 467 (29.3%) | |
| yes | 891 (72.0%) | 234 (65.9%) | 1125 (70.7%) | |
| Missing | 1 | 0 | 1 | |
| **Bleeding** | | | | < 0.001 |
| no | 977 (78.9%) | 87 (24.5%) | 1064 (66.8%) | |
| yes | 261 (21.1%) | 268 (75.5%) | 529 (33.2%) | |

*(Continued)*

**Table 1.** (Continued)

| | Severity 1 (N = 1238) | Severity 2 (N = 355) | Total (N = 1593) | p value |
|---|---|---|---|---|
| **Rash** | | | | 0.002 |
| no | 1176 (96.2%) | 327 (92.1%) | 1503 (95.2%) | |
| yes | 47 (3.8%) | 28 (7.9%) | 75 (4.8%) | |
| Missing | 15 | 0 | 15 | |
| **Leucocytes (WBC)** | | | | < 0.001 |
| Mean (CI) | 5.7 (5.2, 6.2) | 3.8 (3.4, 4.2) | 5.4 (4.9, 5.8) | |
| Range (Min - Max) | 0.1–81.8 | 1.2–7.8 | 0.1–81.8 | |
| Missing | 887 | 277 | 1164 | |
| **Neutrophils (NEU)** | | | | < 0.001 |
| Mean (CI) | 66.9 (65.1, 68.7) | 50.1 (45.9, 54.3) | 63.9 (62.1, 65.6) | |
| Range (Min - Max) | 14.9–91.8 | 14.2–87.3 | 14.2–91.8 | |
| Missing | 887 | 278 | 1165 | |
| **Lymphocytes (LYM)** | | | | < 0.001 |
| Mean (CI) | 19.4 (18.0, 20.8) | 30.7 (27.2, 34.2) | 21.5 (20.1, 22.8) | |
| Range (Min - Max) | 2.2–67.6 | 5.4–66.9 | 2.2–67.6 | |
| Missing | 887 | 278 | 1165 | |
| **Erythrocytes (RBC)** | | | | 0.006 |
| Mean (CI) | 4.5 (4.5, 4.6) | 4.8 (4.6, 4.9) | 4.6 (4.5, 4.6) | |
| Range (Min - Max) | 3.0–7.1 | 3.5–5.9 | 3.0–7.1 | |
| Missing | 887 | 278 | 1165 | |
| **Hemoglobin (Hb)** | | | | 0.007 |
| Mean (CI) | 133.9 (132.3, 135.6) | 139.4 (135.7, 143.2) | 134.9 (133.4, 136.5) | |
| Range (Min - Max) | 76.0–178.0 | 92.0–171.0 | 76.0–178.0 | |
| Missing | 887 | 278 | 1165 | |
| **Hematocrit (HCT)** | | | | 0.014 |
| Mean (CI) | 0.4 (0.4, 0.4) | 0.4 (0.4, 0.4) | 0.4 (0.4, 0.4) | |
| Range (Min - Max) | 0.0–0.5 | 0.3–0.5 | 0.0–0.5 | |
| Missing | 887 | 278 | 1165 | |
| **Platelets (PLT)** | | | | < 0.001 |
| Mean (CI) | 142.4 (135.7, 149.1) | 73.4 (61.4, 85.4) | 130.0 (123.6, 136.4) | |
| Range (Min - Max) | 11.0–371.0 | 7.0–210.0 | 7.0–371.0 | |
| Missing | 887 | 278 | 1165 | |

for severity score 1 and 2, respectively; $P<0.001$, Chi-square test). Surprisingly, rash, a common sign of dengue, was only seen in around 5% of patients.

The range in patient age was similar between females and males and also between severity score 1 and 2. However, the age distribution of females of severity score 1 showed strong signs of bimodality, with a first peak around 25 years and the second one around the age of 55–60 years (see Fig 1). In contrast, the age distributions for disease severity score 2 were similar between males and females. The reason why there is such a pronounced increase in middle-aged women is currently not known.

With respect to self-reported number of days of symptoms before being admitted to hospital, there were no differences between males and females but a clear difference between severity score 1 and 2, with the average number of symptom days being higher in patients with more severe dengue (3.5 vs 4.8 days, $P<0.001$, Welch Two Sample t-test).

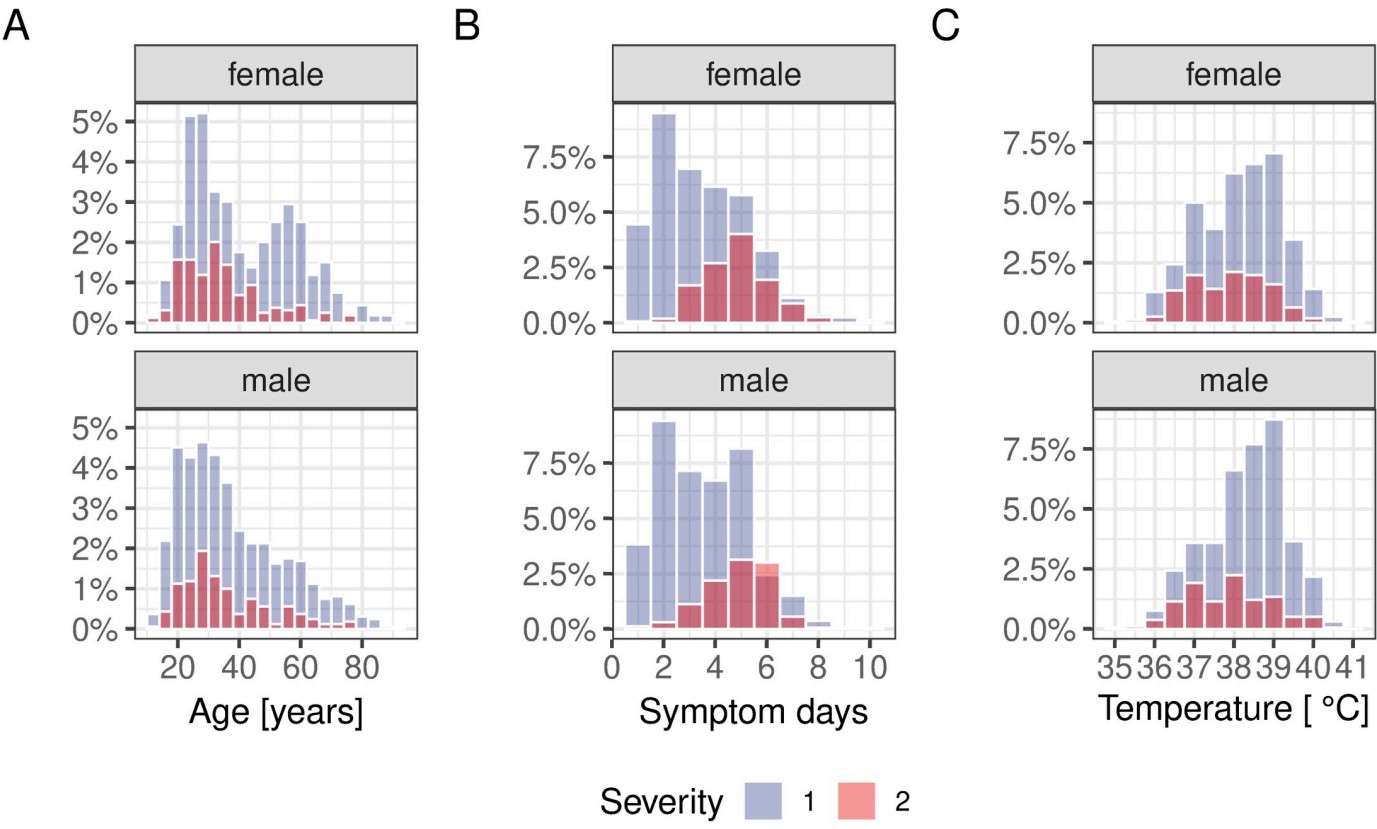

**Fig 1. Age, symptom day and temperature distribution at admission stratified by disease severity.** There are marked differences in the distribution of (A) patient age and (B) pre-hospitalisation periods between disease severity 1 and 2. Further, females showing a pronounced bimodal age distribution, in particular those diagnosed with severity score of 1. (C) The distribution of temperature on admission was similar between men and women but was on average higher in individuals with severity score 1.

The temperature profile differed marginally between severity score 1 and 2. Individuals with a higher severity score had on average a slightly lower temperature on admission than those with severity score 1 (37.9 vs 38.2, $P<0.001$, Welch Two Sample t-test).

## Blood biomarkers of disease

Biomarkers of interest were lymphocyte and neutrophil counts as representatives of both the innate (neutrophils) and the adaptive immune response (lymphocytes) to an infection. As expected, both neutrophil and lymphocyte counts showed significant associations with disease severity (Fig 2A and 2B, respectively). However, individuals with a higher severity score were characterised by lower neutrophil (50% vs. 67%, $P<0.001$, Welch Two Sample t-test) and higher lymphocyte counts (31% vs. 19%, $P<0.001$, Welch Two Sample t-test), leading to an overall lower NLR's in patients with a higher disease severity score (2.7 vs. 6.3, $P<0.001$, Welch Two Sample t-test, Fig 2C).

A similar picture emerges when looking at other blood markers of disease, except erythrocytes (RBC). Individuals with severity score 2 had on average a lower white blood cell (WBC) count (3.8 vs. 5.7, $P<0.001$; Fig 2E) and less platelets (73 vs. 142, $P<0.001$; Fig 2F).

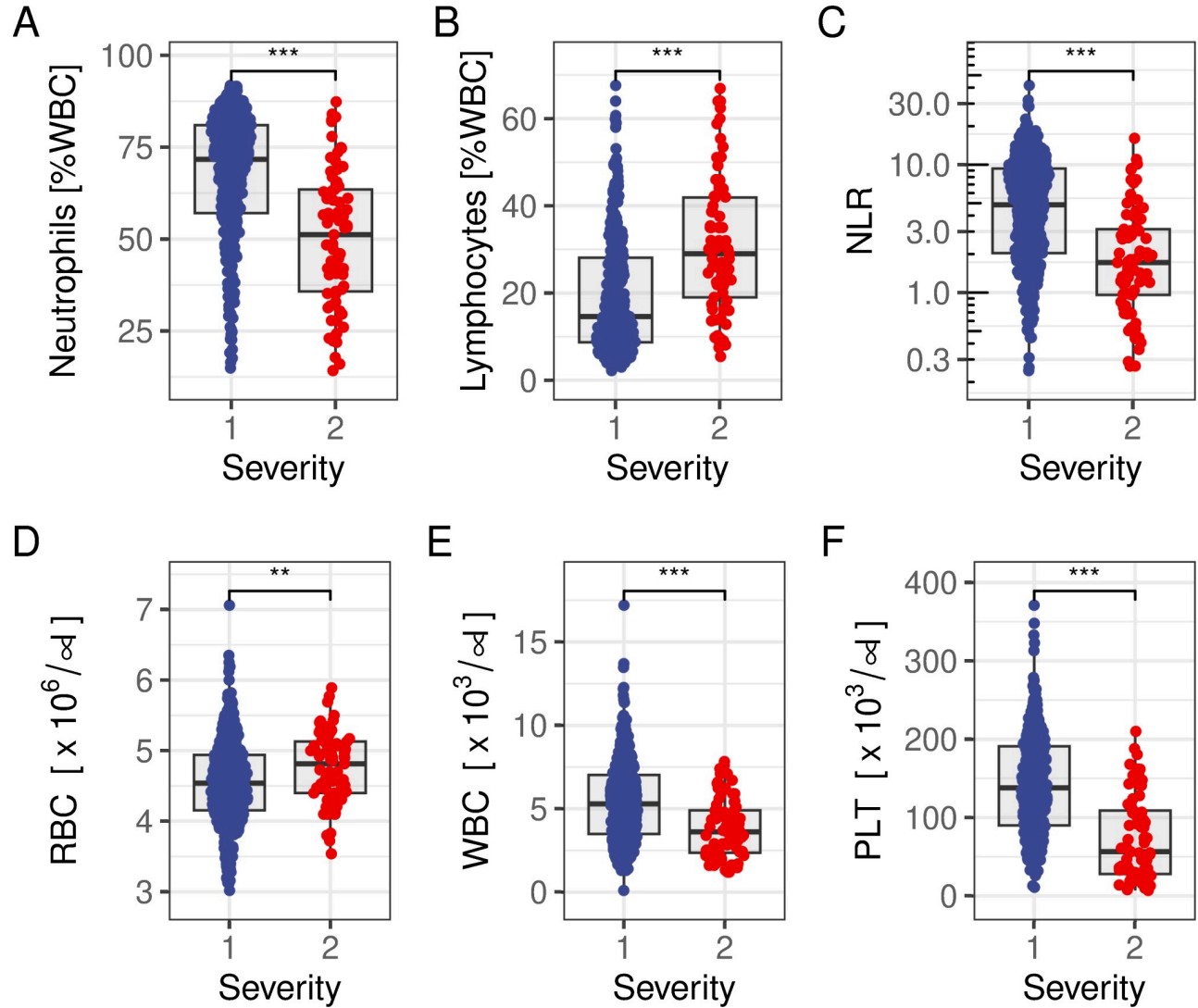

**Fig 2. Distribution of blood markers of disease stratified by severity. (A)** Neutrophils and **(B)** lymphocytes show significant differences between severity score 1 and 2, leading to a negative correlation between NLR and diagnosed severity **(C)**. **(D)** Red blood cell counts (RBC) are similar between individuals with a severity score of 1 and 2, whereas both **(E)** leucocyte (WBC) and **(F)** platelet (PLT) counts show a negative correlation. Statistical significance denoted as *: P<0.05, **: P<0.01, ***: P<0.001. Note, NLR (C) is plotted on a log-scale.

## Hospitalisation length

The next question we addressed was how the length of hospital stay is affected by disease severity and the various markers thereof. The data showed that hospitalisation length was on average lower in patients who were admitted with a higher dengue severity score (4.8 vs. 5.6 days, $P<0.001$, Welch Two Sample t-test; Fig 3A). On the other hand, hospital stay was positively correlated with age, NLR and temperature on admission ($P<0.001$ in all cases, GLM with Poisson error structure, Fig 3B, 3C and 3D).

Of interest, when testing the other infection makers (RBC, WBC, and PLT), all showed a significant association with the length of hospital stay irrespective of dengue severity score ($P<0.05$; $P = 0.02$; $P<0.001$; GLM with Poisson error structure). Taken together, despite the aforementioned negative correlation between dengue disease severity score and hospitalisation

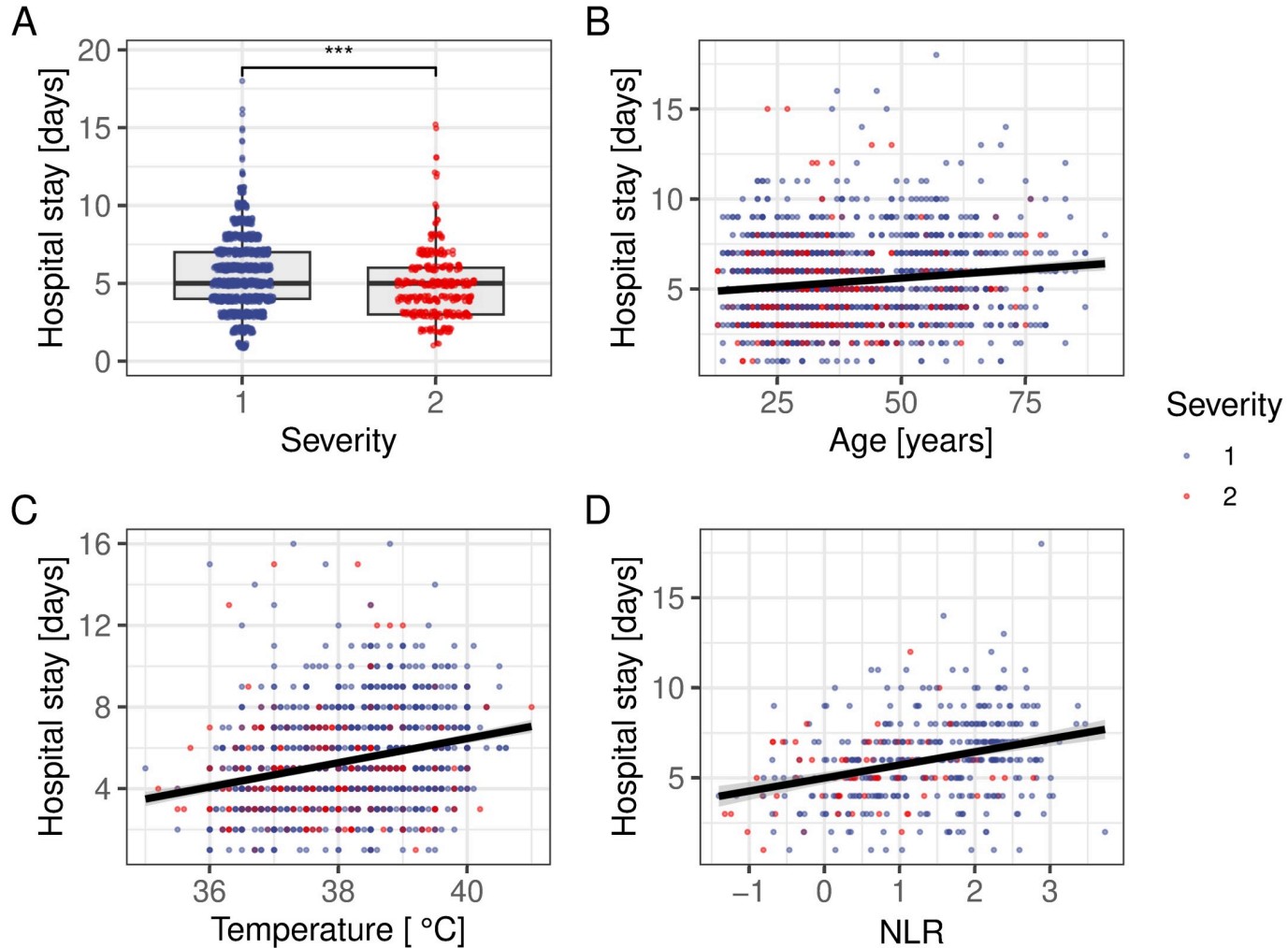

**Fig 3. Markers of prolonged hospital stays. (A)** Disease severity is negatively correlated with the length of stay. **(B)** Patient age, **(C)** temperature on admission and **(D)** neutrophil-to-lymphocyte ratio (NLR) shows a positive relationship with the duration of hospitalisation. Best fit regression lines based on GLMs are shown in black.

length, traditional non-specific infection markers appeared predictive of prolonged hospital stays.

### Symptom days and lead time bias

As shown in Fig 1, there was a significant difference in the average number of symptom days before being admitted to hospital between individuals with a disease severity score of 1 or 2. We next examined how much this influenced the diagnostic picture of patients at the time of hospitalisation admission as well as the length of their subsequent stay.

As demonstrated in Fig 4 (left column), there was a clear negative correlation between diagnostic markers of disease (temperature, WBC, PLT, and NLR) and the number of symptom days prior to hospitalisation. However, when stratified by symptom days we find that the previously observed difference between disease severity disappears. In fact, adding symptom days to the regression model revealed that severity is no longer associated with these markers ($P>0.5$ in all cases, ANOVA), suggesting that the measured responses are driven

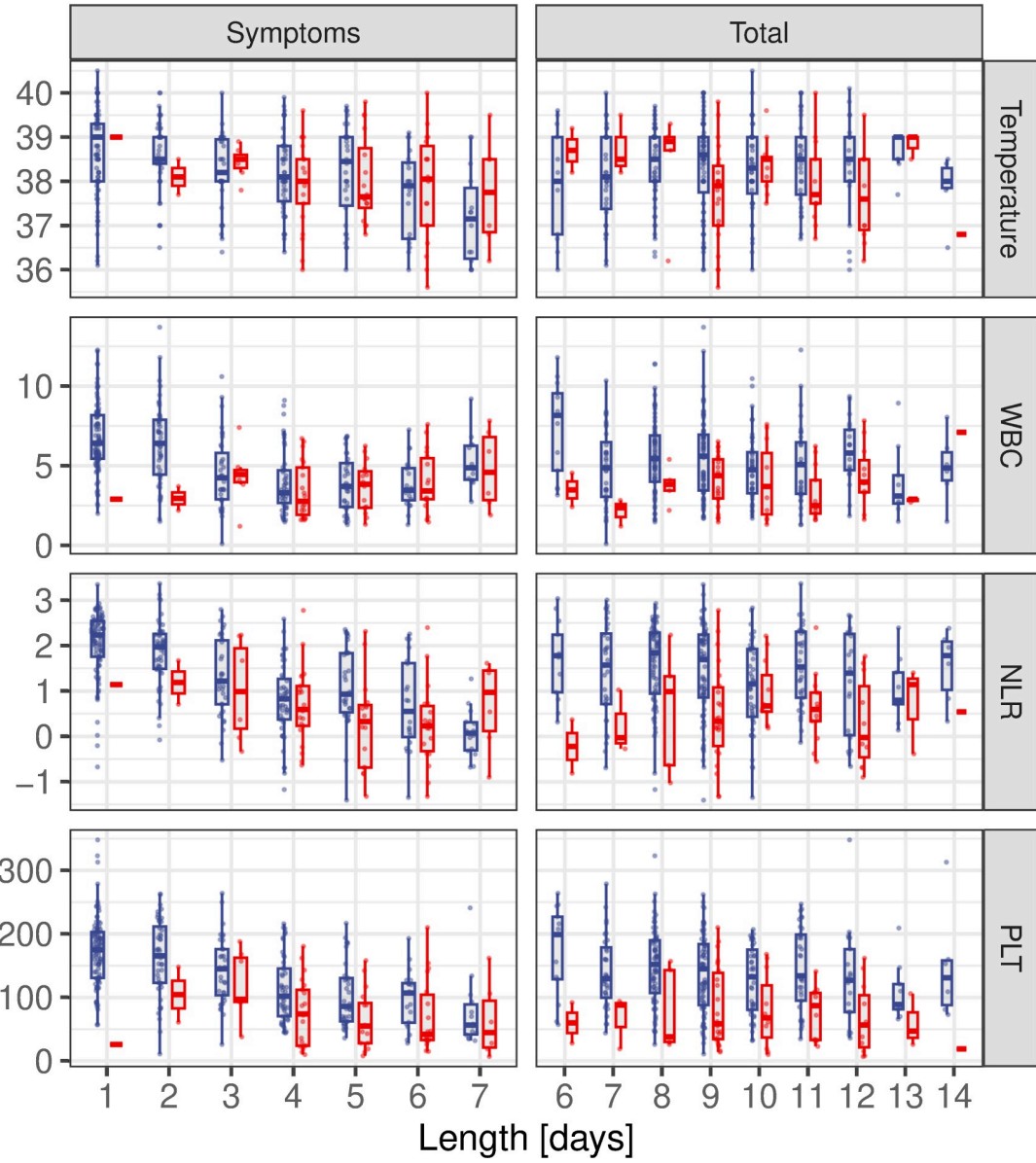

**Fig 4. Severity-stratified diagnostic markers in relation to symptom and total disease days.** All four diagnostic markers (NLR, temperature, leucocytes (WBC) and platelets (PLT)) show a strong, negative correlation with pre-hospitalisation symptom days with little difference between disease severity scores 1 and 2. These correlations disappear when regressing against the total number of disease days, leading to more pronounced differences between dengue severity scores. Note, NLR is plotted on a log-scale to better illustrate the trend.

predominantly by symptom days. Furthermore, we observed a strong negative correlation between pre-and post-admission periods, irrespective of the assigned severity score at admission (S1 Fig), akin to a lead-time bias where early diagnosis associates with prolonged hospitalisation periods.

With this in mind we defined a new variable, *total disease days*, which spans the entire duration from symptom onset to hospital discharge. Plotting the diagnostic markers, taken at the point of admission, against total disease days shows almost no discernible pattern (Fig 4, right column), meaning that their previous associations with hospitalisation length was predominantly driven by how long patients had dengue-specific symptoms for before being admitted to hospital. However, the difference between dengue severity becomes more apparent. Note also, although not all disease markers that associated with severity also showed a trend against the duration of symptoms, they all showed a remarkable stable distribution when stratified by total disease days (S2 Fig).

## Temporal robustness of hospitalisation markers

The importance of considering symptom duration before hospital admission is further confirmed through regression analyses, with the length of stay in hospital as the response variable. The model was run based on all years combined (Fig 5A) and stratified by year (Fig 5B). As shown, age, temperature on admission and symptom days showed the most consistent effect on hospitalisation length. In addition, NLR also appeared to have an effect, with higher values associated with longer stays, but its signal was more uncertain, especially when stratified by year. In fact, statistical inference revealed substantial variations in parameter estimates when comparing different years, implying that the year of sampling can have a critical effect on both the magnitude and even the direction of observed associations between diagnostic markers and length of hospital stay.

## Discussion

Finding reliable markers of hospitalisation periods following a dengue infection, and with it the extent of medical care required, is important for health care management and planning. This is particularly the case in resource limited settings and more so during larger than average outbreaks. The aim of this study was to find minimal but robust signatures to help predict a patient's hospital stay and thus likely health care requirements based on routine clinical diagnostics at time of hospital admission. For this we analysed data of patients admitted with dengue sampled over a period of three years in a number of hospitals in Vietnam.

Previous studies have alluded to the role of various blood biomarkers in differentiating disease severity [7–15] or extended hospitalisation lengths [16–19]. Of note, many of these are unspecific markers of an acute infection that correlate with morbidity or mortality of disease in general, such as leukocytes, neutrophils, or lymphocytes. The ratio of the latter two, the neutrophil-to-lymphocyte ratio, or NLR, has gained much prominence over recent years as a predictive marker of poor infection outcomes for a diverse set of diseases [21,22]. Our study corroborated these associations and demonstrated how an elevated NLR correlates with longer hospital stays. On the other hand, we found some unexpected directionalities with regards to disease severity, here referred to as the assigned severity score at admission, which was negatively correlated with hospitalisation length.

Further analysis taking into consideration the self-reported duration between symptom onset and hospital admission clearly demonstrated how the various markers of disease attenuated with an increased time to admission. That is, the longer the period from symptom onset, the less likely they were found to have elevated temperatures, high levels of neutrophils, or low numbers of leukocytes compared to those who were admitted soon after symptom onset, corroborating previous studies who also highlighted the temporal dynamics of various disease biomarkers since the onset of disease symptoms [12,13,15].

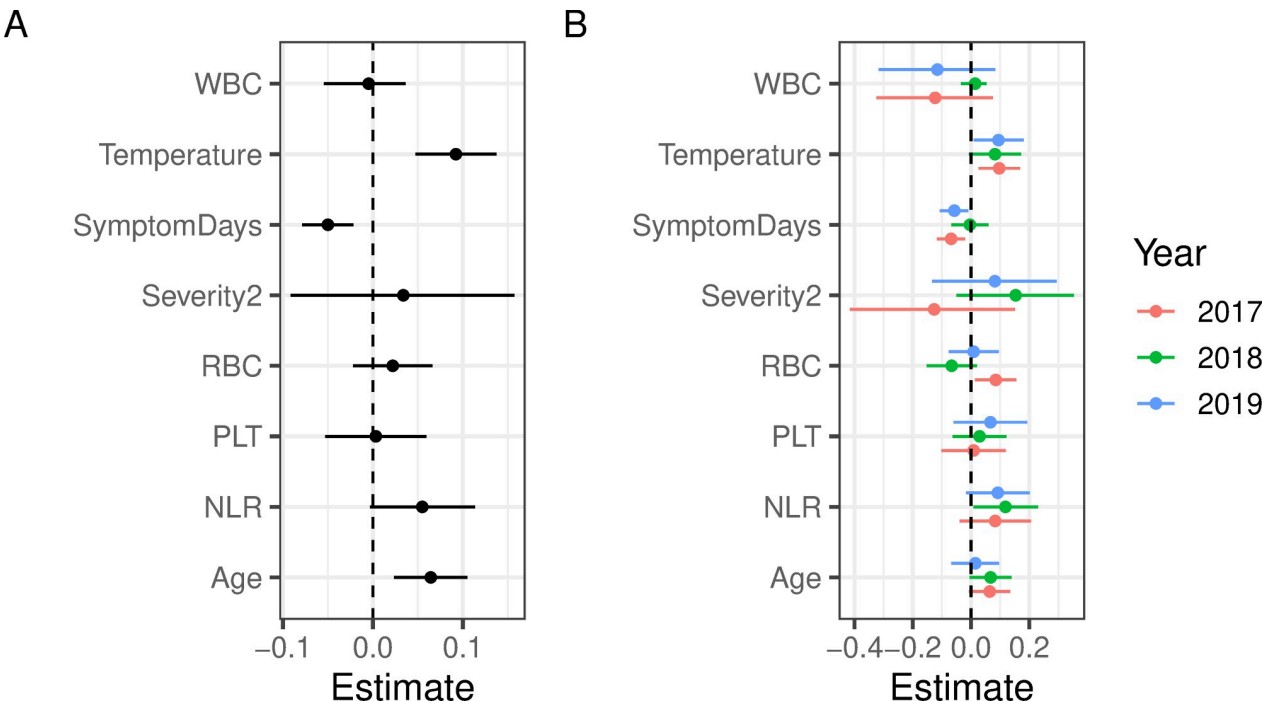

**Fig 5. Importance of disease markers for predicting length of hospitalisation stay.** Estimated means and 95% confidence intervals based on Poisson regression considering all years combined (A) or stratified by year of sampling (B).

An important aspect to consider is that some patients may try to stay home for as long as they can manage and only get hospitalised if more severe disease or complications are suspected, which usually happens in the critical period around day 4–6. As a result, a proportion of patients might get better and never hospitalised, such that those with a higher probability of progressing to more severe disease have longer periods between symptom onset and hospitalisation. On the other hand, patients admitted sooner were found to stay in hospital longer on average than those who were admitted after prolonged symptom periods, and *vice versa*, akin to a lead-time bias. Similar results have previously been reported in a study by Prattay *et al.* [23], who also observed a positive correlation between later hospitalisation and faster recovery time. Taking this into consideration we found that total number of disease days, i.e. the period between symptom onset to discharge, showed a limited association with our selection of biomarkers at the time of admission. Although differences between severity score 1 and 2 were pronounced, these remained stable when plotted against total disease days. What this in turn implies is that these markers are more indicative of the timepoint during the course of the infection than infection or hospitalisation length itself. Note, however, our selection did not include all biomarkers previously put forward as being predictive of dengue severity or infection outcome (e.g. [12,13,24]).

In a predominantly self-limiting infection with very low mortality and no causal therapy, such as dengue, one could expect that the timepoint of hospitalisation has limited influence on the actual length of disease. The window of observation in our study is mainly the time of hospitalisation, but the duration of the pre-hospitalisation phase changes the relative magnitude of biomarkers. It is tempting to speculate that other self-limiting diseases without causal therapies might also be subject to this variant of a lead-time bias.

Based on our analyses, we hypothesise that differences in pre-hospitalisation periods might also account for some of the other reported associations between disease markers and hospital

stays. For example, Lytton *et al.* [25] observed a positive relationship between viraemia at admission and length of stay but also a negative association between symptom days and length of stay, both for primary and secondary infections. In other words, patients admitted to hospital shortly after symptom onset had higher viral loads and longer stays than those who were admitted many days after symptom onset like in our study. The difference between the average number of symptom days before hospitalisation between primary and secondary infections could therefore explain why primary infections were characterised by higher viraemia levels than secondary infections, despite the fact that secondary heterologous infections can be linked with more severe disease and higher viral titres through the phenomenon of ADE [26–29]. It would thus be interesting to re-evaluate other observed differences between primary and secondary infections to understand how much these could be explained simply through differences in admission delays. Unfortunately, in our study we do not know whether patients had their primary or subsequent infection.

Analysis of data sampled over three consecutive years from the same hospitals revealed another important source of uncertainty influencing biomarker discovery. That is, the only risk factor that showed a consistent relationship with a patient's hospital trajectory across all years was their age, which was positively correlated with prolonged stays. Although pre-hospitalisation period, temperature and NLR might also be considered as markers for longer periods until discharge, they showed greater variation between consecutive years than age. More concerning was that the inferred relationships between hospitalisation and other disease markers can swing between positive and negative, raising serious doubts about their use as reliable predictors of health care requirements following dengue hospitalisation. An interesting observation is that the average age of hospital admitted patients was significantly higher in 2017 (37.8 years) than in 2018 (40.1 years) or 2019 (42.6 years). Further studies are required to elucidate how the age distribution itself may influence the inferred relationships between these biomarkers and hospitalisation length.

Hospitalisation costs count for the majority of the health economic burden of dengue, which is a particular problem for low- and middle-income countries [6,30–34]. Understanding, or rather predicting the health care requirements for patients hospitalised with dengue would therefore be of significant benefit in resource limited settings and during large, prolonged epidemic outbreaks. In fact, the data analysed here strongly suggest that the length of hospital stays is subject to resource limitations, with 2017, the year with a large dengue outbreak, having on average the shortest hospitalisation periods. In these cases, having access to reliable indicators of low-risk patients and estimated duration of hospitalisations would help to free up valuable resources for those in greater need.

In summary, our results show that various biomarkers predictive of prolonged hospital stays are strongly influenced by the time between symptom onset and hospital admission. The onset of symptoms as an easily obtainable information can be a valuable asset in avoiding predictive pitfalls. Case-to-case variations, as well as broader shifts of underlying factors, pose the threat of making general predictive factors too unspecific for single patients. Our results further support the notion that dengue is in most cases a self-limiting disease when adequate health care can be provided.

## Supporting information

**S1 Table. Overview of patient characteristics and disease markers at time of hospitalisation.**
(PDF)

**S1 Fig. Length of hospitalisation vs. pre-hospital symptom days.** Duration of hospitalisation is negatively correlated with the reported number of days since symptom onset prior to hospitalisation, irrespective of diagnosed disease severity.
(TIF)

**S2 Fig. Severity-stratified diagnostic markers in relation to symptom and total disease days.** Four diagnostic markers (systolic blood pressure, pulse, haemoglobin (Hb) and haematocrit (HCT)) show different correlations with pre-hospitalisation symptom days with little difference between disease severity scores 1 and 2. Any temporal correlations disappear when regressing against the total number of disease days, leading to more pronounced differences between dengue severity scores.
(TIF)

**S1 Data. Data file containing underlying dengue hospitalisation data analysed in this work.**
(CSV)

## Author Contributions

**Conceptualization:** Mario Recker, Wim A. Fleischmann, Peter G. Kremsner, Thirumalaisamy P. Velavan.

**Data curation:** Trinh Huu Nghia, Nguyen Van Truong, Le Van Nam, Do Duc Anh, Nguyen Trong The, Chu Xuan Anh, Nguyen Viet Hoang, Nhat My Truong.

**Formal analysis:** Mario Recker, Wim A. Fleischmann.

**Funding acquisition:** Le Huu Song, Nguyen Linh Toan, Thirumalaisamy P. Velavan.

**Writing – original draft:** Mario Recker, Wim A. Fleischmann.

**Writing – review & editing:** Mario Recker, Wim A. Fleischmann, Trinh Huu Nghia, Nguyen Van Truong, Le Van Nam, Do Duc Anh, Le Huu Song, Nguyen Trong The, Chu Xuan Anh, Nguyen Viet Hoang, Nhat My Truong, Nguyen Linh Toan, Peter G. Kremsner, Thirumalaisamy P. Velavan.

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
