## [Decision Letter · Decision Letter 0]

5 Aug 2023

Dear Dr Recker,

Thank you very much for submitting your manuscript "Markers of prolonged hospitalisation in severe dengue" for consideration at PLOS Neglected Tropical Diseases. Dengue, the most important arthropod-borne disease, is on the rise. The WHO very recently warned that dengue cases could reach one of the highest rates this year. As the most severe form of the disease can potentially lead to death, it is crucial to understand better the biomarkers of prolonged hospitalisation. Not-needed hospitalization may overwhelm healthcare systems during outbreaks facilitating medical complications and death. In this research, Recker et al studied retrospectively over 2000 individuals hospitalised with dengue in Vietnam for a period of three years (2017-2019). The author's analysis shows that ‘time since symptom onset’ is one of the strongest predictors of hospitalisation length regardless of the severity of dengue illness.

As with all papers reviewed by the journal, your manuscript was reviewed by members of the editorial board and by several independent reviewers. In light of the reviews (below this email), we would like to invite the resubmission of a significantly-revised version that takes into account the reviewers' comments. We cannot make any decision about publication until we have seen the revised manuscript and your response to the reviewers' comments. Your revised manuscript is also likely to be sent to reviewers for further evaluation.

Sincerely,

Daniel Limonta, MD, PhD

Academic Editor

Elvina Viennet

Section Editor

Reviewer's Responses to Questions

**Key Review Criteria Required for Acceptance?**

**Methods**

-Are the objectives of the study clearly articulated with a clear testable hypothesis stated?

-Is the study design appropriate to address the stated objectives?

-Is the population clearly described and appropriate for the hypothesis being tested?

-Is the sample size sufficient to ensure adequate power to address the hypothesis being tested?

-Were correct statistical analysis used to support conclusions?

-Are there concerns about ethical or regulatory requirements being met?

Reviewer #1: The manuscript by Recker et al on markers of prolonged hospitalization is a very interesting manuscript with very important data. I wish to make the following comments

Methods:

1. To analyse the risk factors associated with duration of hospitalization, did the authors consider to evaluate presence of comorbidies: diabetes, obesity etc… Was disease severity at time of admission to hospital recorded and analysed? i.e. how many had dengue with warning signs or DHF when they were admitted?

2. As the readers are not familiar with disease severity score of 1 to 3, can a brief summary be given? Otherwise its very difficult to interpret and understand the data

3. The number of patients classifies has disease severity score of 1 and 2 is less than the total number of patients. The numbers don’t add up.

Reviewer #2: See below

Reviewer #3: (No Response)

**Results**

-Does the analysis presented match the analysis plan?

-Are the results clearly and completely presented?

-Are the figures (Tables, Images) of sufficient quality for clarity?

Reviewer #1: Results

1. the overview of patient characteristics does not provide much data. What about the laboratory parameters? Also could the authors list the symptoms and the proportion of symptoms at the time of presentation

2. 63% of patients having bleeding manifestations is alarming. Rather than the %, can the actual numbers be given. The numbers don’t add up in many parts of the manuscript. 

3. The patients withs severe disease had longer hospitalization? Is this because those who presented late to hospital, had a delay in management (i.g. fluid therapy) and therefore, already had severe disease at the time of hospitalization. It is important to provide clinical disease severity at presentation, as it is not possible to make sense of the data.

Reviewer #2: See below

Reviewer #3: (No Response)

**Conclusions**

-Are the conclusions supported by the data presented?

-Are the limitations of analysis clearly described?

-Do the authors discuss how these data can be helpful to advance our understanding of the topic under study?

-Is public health relevance addressed?

Reviewer #1: General comments: there seem to be quite a bit of missing data due to the retrospective nature of the study. It would be important to analyse the risk factors for prolonged hospitalization and this has not been addressed in this study adequately. As a result of this, many of the interpretations of results, don’t seem to be rational.

Reviewer #2: See below

Reviewer #3: (No Response)

**Editorial and Data Presentation Modifications?**

Reviewer #1: (No Response)

Reviewer #2: N/A

Reviewer #3: (No Response)

**Summary and General Comments**

Reviewer #1: General comments: there seem to be quite a bit of missing data due to the retrospective nature of the study. It would be important to analyse the risk factors for prolonged hospitalization and this has not been addressed in this study adequately. As a result of this, many of the interpretations of results, don’t seem to be rational.

Reviewer #2: The authors report their analysis of a dataset collected over several years in Vietnam looking at risk factors/prognostic markers that may determine length of hospital stay and potentially be useful for healthcare planning/resource allocation. Their main finding is of an “unexpected correlation between markers of disease severity and hospitalisation length, which can be resolved by taking the period of dengue symptoms prior to hospitalisation into account”. However this is not actually unexpected!

The first major point is that the natural evolution of dengue signs, symptoms and laboratory results is well known and many observational/descriptive studies and clinical trials already take the period of time from illness onset into account – with analyses described by fever day, illness day etc. The quite extensive existing literature should be acknowledged and summarised. Specifically with respect to viremia and primary/secondary infections (as mentioned in the discussion) there are several publications, including a very large dataset (Vuong NL et al. CID, 2021) that look at these data by day of illness and show relationships with markers of clinical severity. 

The value of the work presented is in corroborating this well-known fact and showing mathematically that taking the parameter referred to here as “symptom day” into account has a major influence on the various biomarkers assessed. 

The second major point is that the authors cannot really comment on relationships with severity when they exclude all death cases and those with severity score 3. The remaining patients are effectively those with and without warning signs, and since there is a lot of missing data even within these groups it is not surprising that relationships to severity were difficult to identify. Please include a section on the quality of the data (degree of missingness) in the main text and comment on how this might have influenced the results. Also comment on the exclusion of all severe cases….. 

Many factors other than clinical severity determine when an individual is hospitalised (some of which are mentioned in the text), but also when they are discharged. Discharge guidelines (often including arbitrary lab values rather than focusing on clinical severity markers), the need for beds for other patients, the day of the week (staff are often unavailable at weekends to complete discharge papers so discharges are more common on Fridays and Mondays) etc. etc. A more detailed discussion of the factors that may affect both admission and discharge days is warranted, as well as mention of the limited utility of length of hospitalisation as an indicator of clinical severity. 

It is not clear to me whether all the clinical/lab data analysed was from the first assessment only? Please clarify exactly which data are included. Also for the outcome severity scoring please include more specific details of how this was done, potentially in the appendix. How many assessments were required to be able to give a score to an individual? What happened if key variables were missing?

The author summary refers to a “delay in admission in those patients with higher severity scores”. This interpretation of the data, and specifically use of the word “delay” is unwarranted and potentially problematic for health services in endemic settings. There are many reasons why slightly more symptomatic cases may be overrepresented among the later admissions, but there is nothing to suggest that the outcome would have been any different if they had been admitted earlier. We know that a huge proportion of mildly symptomatic dengue cases never present to clinical services at all – maybe the more stoical among them were feeling better by day 3/4/5, meaning that the majority of those who presented to a health facility at this time were those who felt a bit worse or were more worried than their counterparts. Suggesting that all these individuals should have been admitted earlier could increase the burden on health services

Reviewer #3: General: 

This is a well written manuscript on an important topic. The data set used seems to be promising and the methodology is sound. However, there seems to be a lack of reference to the natural history of the disease and its features over time – for example the onset of the ‘critical period’ at around day of illness 4-6 (with more severe disease for a subset of Dengue patients). 

It seems that the authors conclude – as one of their findings - that it is important to take into account ‘total illness days’ (the authors also call this a ‘lead time bias’). It is well known in the field that ‘day of illness’ is an important variable for the clinical evaluation of dengue patients and that depending on this, the presence or absence of certain clinical signs and symptoms (e.g., ‘warning signs’) has to be interpreted differently. Thus, adjusting for ‘day of illness’ is not a new finding and results that don’t take into account ‘day of illness’ might be misunderstood. This leads to a situation where the findings of this manuscript don’t integrate well with the body of literature. This reviewer believes that if the analysis would take into account (adjust for/stratify by) ‘day of illness’ from the very start, the findings would be much easier to interpret. 

In many countries in Southeast Asia (presumably also in Vietnam), patients try to stay at home as long as they can manage and are only hospitalized for dengue if more severe disease or complications are suspected, either as a result of dengue itself (which usually happens around day 4-6, in the ‘critical period’) or as a result of existing comorbidities or difficulties at home (being alone, living too far from a health facility). This means that around day of illness 3-5, a proportion of the patients with symptomatic Dengue actually get better and are never hospitalized. The ‘pool’ of patients that proceeds to more severe disease is smaller starting at around day of illness 3-5, but it includes the patients that have a higher probability of more severe disease. 

I hope that these introductory remarks help and below are more specific comments for the authors. I would recommend major revisions. 

Background: 

- Line 100: “Our analysis reveals…” – should this not go into the results?

Methods: 

- Can you please include how dengue was confirmed by laboratory diagnosis?

- Line 109: Why was only the first blood draw analysed per patient? Were repeated blood results available per patient? What is the day of illness distribution at time of hospitalization / enrolment? Can you please include this information into table 1? Later in the manuscript (caption figure 4) it becomes clear that diagnostic markers show a strong correlation with day of illness…

- Line 113: What was the justification to restrict the analysis to the variables described at this point? Was there an a-priori analysis plan? Who decided on this set of variables?

- Line 122/123: Can you provide more information about the adaptation of the WHO severity score by the Vietnamese Ministry of Health? This could go into the appendix. 

Table 1: 

- 75% of the data comes from 2017. Can you stratify outcomes by year in table 1? Was a heterogeneity assessment conducted?

Results: 

- Line 151: The authors report about a potential non-linear relationship of age with the outcome. At a later point, age was modelled linear. Were you considering introducing splines for age?

- Lines 157ff: See my explanations above about the fact that the patients that get hospitalized at around the critical period (day of illness 4-6) have a higher baseline probability of severe disease. 

- Line 174: Because of the natural history of disease, blood biomarkers should really be presented by ‘day of illness’ (‘or day of illness bins’ if necessary for sample size reasons). 

- Line 174ff: It seems the results of the blood biomarkers are presented as univariate. Would it be possible to present a regression with severity score 2 as the outcome?

- Line 199: Could it be that the result that ‘hospitalization length was lower in patients with higher severity score’ is due to the fact that they were ‘further along’ in their natural history of disease, being hospitalized on a later day of illness? The finding itself needs to be interpreted in the context. 

- Line 207: Again, the “traditional non-specific infection markers” should be analyzed adjusting for day of illness.

- Line 228: “early hospitalization” rather than “early diagnosis”?

- Line 246ff.: Maybe the different severity mix including the distribution of day of illnesses is partially responsible for the heterogeneity by year shown in figure 5 and the accompanying text?

Figure 3: 

- As mentioned before, it might be interesting to see if there are non-linear trends with regard to age, using splines?

Figure 5: 

- This is an important figure, showing the heterogeneity between years. Would it be good to talk about this heterogeneity in the methods section already, see my comment for table 1?

Discussion: 

- Line 276 and line 285: These results are likely to be due to the natural history of disease, which calls for stratifying or adjusting for ‘day of illness’ at time of hospitalization. See my general comments above. This theme is reiterated…

- Line 328: The authors mention for the first time that the year 2017 had a large dengue outbreak with on average shorter hospitalization periods. The authors speculate if the hospital capacity was more strained in 2017 and therefore people were discharged earlier than in subsequent years? If yes, this would question their findings substantially as their results are driven by the 75% of patients from 2017. It might be important to do a sub-analysis of 2017 only data to confirm if the trends of the results reported are valid!

PLOS authors have the option to publish the peer review history of their article (what does this mean?). If published, this will include your full peer review and any attached files.

Reviewer #1: No

Reviewer #2: No

Reviewer #3: No

Figure Files:

Data Requirements:

Please note that, as a condition of publication, PLOS' data policy requires that you make available all data used to draw the conclusions outlined in your manuscript. Data must be deposited in an appropriate repository, included within the body of the manuscript, or uploaded as supporting information. This includes all numerical values that were used to generate graphs, histograms etc.. For an example see here: http://www.plosbiology.org/article/info:doi%2F10.1371%2Fjournal.pbio.1001908#s5.
---

## [Editor Report · Decision Letter 1]

15 Jan 2024

Dear Dr Recker,

We are pleased to inform you that your manuscript 'Markers of prolonged hospitalisation in severe dengue' has been provisionally accepted for publication in PLOS Neglected Tropical Diseases. The authors have properly discussed and addressed the reviewers’ comments and suggestions. Furthermore, the limitations of the study were fairly covered.

Large outbreaks of dengue involve a significant burden on the healthcare systems of developing countries. This is why the appropriate allocation of limited resources is critically important. Recker et al. analyzed dengue hospitalization data of over 2000 Vietnamese patients over three years and found a negative correlation between dengue severity and length of hospitalization. This finding, along with other analyzed factors, may be useful for healthcare planning and resource allocation.

Best regards,

Daniel Limonta, MD, PhD

Academic Editor

Elvina Viennet

Section Editor

---

## [Editor Report · Acceptance letter]

18 Jan 2024

Dear Dr Recker,

We are delighted to inform you that your manuscript, "Markers of prolonged hospitalisation in severe dengue," has been formally accepted for publication in PLOS Neglected Tropical Diseases.

Best regards,

Shaden Kamhawi

co-Editor-in-Chief

Paul Brindley

co-Editor-in-Chief
